# *MAT1-1* and *MAT1-2 Ophiocordyceps xuefengensis* and Comparison of Their Chemical Composition

**DOI:** 10.3390/biology13090686

**Published:** 2024-09-02

**Authors:** Juan Zou, Yating Zhang, Yan Luo, Miaohua Fu, Beilin Sun, Shenggui Liu

**Affiliations:** 1Key Laboratory of Research and Utilization of Ethnomedicinal Plant Resources of Hunan Province, Huaihua University, Huaihua 418000, China; swlsg@163.com; 2Hunan Provincial Higher Education Key Laboratory of Intensive Processing Research on Mountain Ecological Food, College of Biological and Food Engineering, Huaihua University, Huaihua 418000, China; 3College of Biological and Food Engineering, Huaihua University, Huaihua 418000, Chinasbl2200120525@outlook.com (B.S.)

**Keywords:** *Ophiocordyceps xuefengensis*, mating-type genes, loop-mediated isothermal amplification, chemical composition

## Abstract

**Simple Summary:**

Sexual reproduction in *Cordyceps sensu lato* is regulated by the mating-type locus, which has two alternate idiomorphs: *MAT1-1* and *MAT1-2*. *Ophiocordyceps xuefengensis*, a heterothallic *Cordyceps* used as a traditional tonic by the Yao ethnic group in southern China, produces sterile fruiting bodies regardless of whether it is the *MAT1-1* or *MAT1-2* strain. However, the impact of mating types on the quality of these fruiting bodies is not well understood. In this study, we developed a loop-mediated isothermal amplification (LAMP) method that quickly and efficiently distinguishes between *MAT1-1* and *MAT1-2* strains of *O. xuefengensis*, with advantages including a short duration (1 h or less), visual detection, and the elimination of the need for a thermal cycler and gel electrophoresis. We also systematically compared biochemical contents of the fruiting bodies from *MAT1-1* and *MAT1-2* strains. Our findings show that the *MAT1-1* strain of *O. xuefengensis* is a promising alternative for natural *Cordyceps production*, as it contains higher levels of adenosine and essential amino acids, as well as lower levels of toxic elements in the fruiting bodies compared to the *MAT1-2* strain. This study provides a new approach to addressing issues related to unstable artificial cultivation, long cultivation periods, and high heavy metal content in *Cordyceps*.

**Abstract:**

Many *Cordyceps sensu lato* species are used as traditional Chinese medicines. However, *Cordyceps* are entomopathogenic fungi in the family Clavicipitaceae of Ascomycota, and excessive harvesting severely disrupts natural habitat ecosystems. Artificial cultivation of *Cordyceps* fruiting bodies offers a viable strategy to protect the ecological environment and mitigate the depletion of wild resource. In this study, mononucleate hyphae were selected using DAPI fluorescence staining, the *MAT1-1* and *MAT1-2* strains of *O. xuefengensis* were successfully distinguished using loop-mediated isothermal amplification (LAMP). The chemical composition and bioactive components of fruiting bodies produced by these strains were compared. Results showed that the levels of adenosine, thymidine, adenine, guanosine, uridine, total amino acids, and total essential amino acids in the fruiting bodies of *MAT1-1* strains were 1.31 mg/g, 0.15 mg/g, 0.26 mg/g, 2.40 mg/g, 2.34 mg/g, 270.3 mg/g, and 102.5 mg/g, respectively, which were significantly higher than those in the *MAT1-2* sample. Contrastingly, the fruiting bodies of *MAT1-2* strains contained higher levels of mannose and polysaccharides, at 11.7% and 12.2%, respectively. The levels of toxic elements such as Al, Pb, As, and Hg in the *MAT1-1* fruiting bodies were 1.862 mg/kg, 0.0848 mg/kg, 0.534 mg/kg, and 0.0054 mg/kg, respectively, which were markedly lower than those in the *MAT1-2* fruiting bodies.

## 1. Introduction

Many *Cordyceps sensu lato* species, such as *Ophiocordyceps sinensis*, *Isaria cicadae*, and *Cordyceps militaris*, are used as valuable traditional medicines or novel food in China, Japan, Korea, and other Eastern Asian countries [1,2,3,4]. Numerous studies have shown that *Cordyceps* contains a variety of bioactive constituents, including polysaccharides, adenosine, cordycepic acid, cordycepin, ergosterol, and minerals. Liu et al. [5] revealed that the polysaccharide from *O. sinensis* activated RAW264.7 macrophage cells and exhibited a stronger immunomodulatory effect through the phosphatidylinositol 3-kinase/protein kinase B (PI3K/Akt), extracellular signal-regulated kinase (ERK), c-Jun N-terminal kinase (JNK), and p38 mitogen-activated protein kinase (MAPK) signaling pathways. Le et al. [6] demonstrated that the polysaccharide PS-T80 from *O. sobolifera* exhibits promising antioxidant activity, with a scavenging rate of 93.85% for 1,1-Diphenyl-2-picrylhydrazyl (DPPH) at a concentration of 2.5 mg/mL. Adenosine, a major nucleoside in *Cordyceps*, can regulate macrophage phenotypic switching to reduce inflammatory responses. It also serves as a chemical marker for quantitatively monitoring the product quality of *O. sinensis* and *C. militaris*, as well as their substitutes in Asia [7,8]. Wang et al. [9] reported that cordycepic acid significantly inhibited the proliferation of human lung cancer A549 cells by regulating the Nrf-2/HO-1/NLRP3/NF-κB signaling pathway. Deng et al. [10] discovered that cordycepin lowered the expression of phagocytosis immune checkpoint cluster of differentiation 47 (CD47) in tumor cells, including the murine colon cancer cells CT26, human T-lymphoblastic (Jurkat) cells, and human colon cancer cell lines SW48, thereby promoting the phagocytosis of tumor cells by macrophages and enhancing anti-tumor immunity. Additionally, cordycepin exhibits excellent protective effects against inflammatory damage from acute lung injury (ALI), allergic asthma, liver fibrosis, rheumatoid arthritis, myocardial ischemia-reperfusion injury, and atopic dermatitis [11,12,13,14]. Ergosterols, a class of unique steroidal compounds specific to fungal cell membranes, belong to the group of organic compounds known as ergosterols and their derivatives. They consist of one cyclopentane ring, one cyclohexane ring, one cyclohexene ring, and one cyclobutane ring in their structure, and have been reported to exhibit antitrypanosomal, antimicrobial, antitumor, and various other activities [15,16,17].

*Ophiocordyceps xuefengensis*, the sister taxon of *O. sinensis*, parasitizes the larvae of *Phassus nodus* (Hepialidae) found in the living roots or trunks of *Clerodendrum cyrtophyllum* in the Xuefeng Mountains of Hunan Province in south China, and is well known as a folk tonic medicine by the local Yao ethnic group, also regarded as an ideal alternative for *O. sinensis* [18,19]. *O. xuefengensis* has been shown to be rich in various active chemical compounds, including nucleosides, nucleobases, cordycepic acid, amino acids, fatty acids, and ergosterol [20,21,22]. Qin et al. [23] demonstrated that the *O. xuefengensis* exhibited higher antioxidant capability than both *C. militaris* and *O. sinensis*. Jin et al. [24] indicated that the water extract of *O. xuefengensis* exerted antitumor activity against Michigan Cancer Foundation-7 (MCF-7), hepatocellular carcinoma G2 (HepG2), prostate cancer-3 (PC-3), A549, and Raji cell lines in vitro. Zheng et al. [25] demonstrated that the aqueous extract of *O. xuefengensis* can enhance antitumor activity against human HepG2 liver cancer by promoting the proliferation of dendritic cell-cytokine-induced killer (DC-CIK) cells.

With the rising market demand for this medicinal fungus, the prices of natural *Cordyceps* have soared dramatically over the past few decades, leading to excessive harvesting that has endangered the species and severely damaged the ecological systems of their natural habitats in China [2]. Artificial cultivation of *Cordyceps* fruiting bodies offers a viable strategy to safeguard the ecological environment and address the constraints of wild resource depletion. Sexual reproduction is crucial for the sustainable development of the *Cordyceps* industry, as mating-type loci play a decisive role in determining gender and sexual reproduction. In filamentous ascomycetes, sexual reproduction is governed by a single locus known as the mating type (*MAT*) locus. This locus has two highly divergent and nonhomologous variants, termed *MAT1-1* and *MAT1-2*, which are referred to as idiomorphs rather than alleles. Heterothallic (self-sterile) fungal strains contain only one single idiomorph and require mating with a strain of the opposite mating type to achieve successful ascus formation [26]. In some heterothallic species of *Cordyceps sensu lato* in ascomycetes, such as *C. militaris* and *O. xuefengensis*, strains containing either the *MAT1-1* or *MAT1-2* idiomorph are self-sterile; however, they can still produce sterile fruiting bodies [26,27]. It remains unclear, however, whether the mating types of *Cordyceps* species influence the quality of these fruiting bodies.

The morphological differentiation between the *MAT1-1* and *MAT1-2* strain without quarantine status is difficult, thus rapid on-site identification is needed. An excellent approach is the loop-mediated isothermal DNA amplification (LAMP) assay, which has recently been shown to be an efficient and convenient technology for identifying viruses, bacteria, fungi, and other pathogens [28,29,30]. LAMP is extremely sensitive and highly specific because it uses two primer sets to effectively identify target DNA sequence sites [28]. Due to the DNA strand displacement activity of the *Bst* DNA polymerase, LAMP reactions are carried out at a constant temperature, eliminating the need for a thermal cycler compared to conventional polymerase chain reaction (PCR)-based assays. Moreover, LAMP has a distinct advantage over PCR-based technologies in its resilience against latent inhibitors in DNA extract, thereby eliminating the need for DNA purification [31]. Because of its simplicity, speed, and lack of requirement for a thermal cycler, LAMP can be conducted on-site using portable constant-temperature real-time detection equipment.

In this study, we developed a novel method for rapidly distinguishing between the *MAT1-1* and *MAT1-2* strains of *O. xuefengensis* using the LAMP reaction. This technique has several advantages, including the elimination of DNA purification steps, providing visual results within 1 h or less, and bypassing the need for a thermal cycler and gel electrophoresis. Furthermore, we conducted a thorough analysis comparing the contents of polysaccharides, adenosine, cordycepic acid, cordycepin, amino acids, fatty acid, and element components between the *MAT1-1* and *MAT1-2* fruiting bodies and investigated the effects of the mating-type genes on these components.

## 2. Materials and Methods

### 2.1. Materials

Mononucleate hyphae were selected for fruiting body cultivation using DAPI (4′,6-diamidino-2-phenylindole) staining: conidia from 3-week-old colonies of *O. xuefengensis* were cultured as reported by Tkaczuk and Majchrowska-Safaryan [32], and the hyphae were grown on a cover glass until reaching the appropriate growth stage. Methanol at 4 °C was then applied to the side of the cover glass containing the hyphae and left for 10–15 min. Subsequently, the hyphae were washed three times with phosphate-buffered saline (PBS) at pH 7.4, with each wash lasting 5 min and involving occasional gentle shaking. Next, 0.4% Triton X-100 was added to the cover glass containing the cultures and left for 5–10 min. The cultures were then washed with PBS 2–3 times, each wash lasting for 5 min. Following this, the cultures were treated with 1 μg/mL DAPI staining solution for 10 min, washed with PBS 2–3 times (each wash lasting 5 min), and then examined using laser confocal microscopy (Sunny Optical Technology Co., Ltd., Ningbo, China). Mononucleate hyphae of *O. xuefengensis* were used in culture to cultivate both liquid mycelium and fruiting bodies. The fungal strains numbered 1–14 were isolated from the fresh stromata of *O. xuefengensis* collected from the Xuefeng Mountains in China.

The strain ZJ3, genotype *MAT1-2*, is preserved in the China Center for Type Culture Collection (no. CCTCC M 2015777). The strain ZJ1 has the genotype *MAT1-1* (GenBank no. MH176300). Strain no. 1–14, collected from Xuefeng Mountain at an altitude of over 400 m, with coordinates ranging from 110.40 to 110.69° E and 27.06 to 27.23° N, has ITS sequences with GenBank no. PQ159738–PQ159751, respectively. The fruiting bodies of *O. xuefengensis* were cultivated artificially at 19 °C in dark conditions on solid media as reported by Zou et al. [22]. The media comprised 40 g rice, 20 g oat, 10 g millet, 2 g pupa powder, 0.4 g CaCO_3_, and 0.4 g CaSO_4_·2H_2_O, along with a nutrient solution (120 mL) containing 200 g/L potato infusion, 10 g/L sugar, 5 g/L peptone, 1.18 g/L Na_2_HPO_4_, 1.13 g/L KH_2_PO_4_, 1.5 g/L MgSO_4_·7H_2_O, and 2 g/L KNO_3_ [22]. These components were mixed and poured into a 31.0 cm × 21.0 × 11.5 cm polyethylene cultivation pot. The state of fruiting bodies was captured using a Canon camera with an EF-S 55-250 mm lens (Canon, Zhuhai, China). The fruiting bodies of natural *O. xuefengensis* were derived from the larvae of *Phassus nodus* (Hepialidae) found in the living roots or trunks of *Clerodendrum cyrtophyllum* in the Xuefeng Mountains of Hunan Province, southern China. The *MAT1-1* and *MAT1-2* fruiting bodies were artificially cultivated on solid media in the laboratory, as detailed in our materials.

### 2.2. LAMP Reaction

The specific primer sets, shown in Table 1, were designed on the basis of conserved domain sequences of the *MAT1-1-1* (MH176302.1) and the *MAT1-2-1* (MH176301.1) of *Ophiocordyceps xuefengensis*, respectively, *using* Primer Explorer version 5 software (available at http://primerexplorer.jp/lampv5e/index.html, accessed on 20 August 2020) from the Eiken Chemical Company (http://primerexplorer.jp/lampv5e/index.html, accessed on 20 August 2020).

A 100 mg fresh sample or a 50 mg dry sample of *O. xuefengensis* was placed in a mortar and mixed with three times its weight of sterile quartz sand. The mixture was thoroughly ground with a pestle, transferred into 300 μL of sterile double-distilled water in a 1.5 mL centrifuge tube, and incubated in a boiling water bath for 10 min. The supernatantcontaining DNA was then collected via centrifugation at 9800× *g* for 5 min, stored at −20 °C, and used within 48 h.

The LAMP reaction was performed in 20 µL mixtures containing 8 mM MgSO_4_, 1.4 mM dNTP mix, 1 × ThermoPol buffer (0.1% Triton X-100, 2 mM MgSO_4_, 20 mM Tris–HCl, 10 mM (NH_4_)_2_SO_4_, 10 mM KCl, pH 8.8), 1.25 µM of each FIP and BIP, 0.125 µM of each F3 and B3, 12 U Bst DNApolymerase (NEB, Ipswich, MA, USA), 1 µL DNA template, 120 µM hydroxynaphthol blue (HNB), 0.8 M betaine, and ddH_2_O to a final volume of 20 µL. The reaction was incubated at 61 °C for 1 h and then terminated by heating at 80 °C for 10 min. DNA amplification was assessed using two methods: 2% agarose gel electrophoresis and visual detection of the color changes in the LAMP mixture with HNB dye.

### 2.3. Analysis of Nucleosides, Cordycepic Acid, and Amino Acid

According to Zou et al. [22], nucleoside analogues and nucleobases were extracted and detected using reverse-phase HPLC with a Waters e2695 HPLC system equipped with a Diamonsil C18 (2) column (Diamonsil, 250 mm × 4.6 mm, 5 µm). The mobile phase was ultrapure water (A) and methanol (B). Elution conditions: 0–3 min at 12% B, 3–3.5 min gradient from 12% to 21% B, 3.5–8.5 min at 21% B, 8.5–9 min gradient from 21% to 35% B, and 9–15 min at 35% B. This was followed by a 40 min flush with 100% B and 30 min reconditioning at 12% B. The flow rate was 1 mL/min, the injection volume was 20 µL, and detection was performed at 260 nm and a temperature of 30 °C. The content of cordycepic acid (D-mannose) was determined via KIO_4_ reaction using a corresponding chemical standard [33]. Ergosterol in the fruiting bodies of *O. xuefengensis* was detected using a Waters e2695 HPLC system with a solvent system of methanol (98%) and water (2%) at a flow rate of 1 mL/min and a total analysis time of 15 min, as reported by Guo et al. [34]. Ergosterol separation was achieved on a C18 column (Diamonsil, 250 mm × 4.6 mm, 5 µm), and detection was performed at 283 nm UV.

The acid hydrolysis and alkaline hydrolysis of the materials were performed to quantitatively analyze amino acids [35]. The hydrolyzed amino acid samples were filtered through a 0.45 µm membrane. Automated pre-column derivatization of the filtrate was achieved using o-phthalaldehyde (OPA) and 2-mercaptoethanol for primary amino acids, followed by 9-fluorenylmethyl chloroformate (FMOC-Cl) for secondary amino acids. Separation was performed using a pre-packed Hypersil ODS C18 column (4.6 × 250 mm, particle size 5 µm; Dalian Elite Analytical Instruments Co., Dalian, China), and determination was carried out with a photometric detector at 338 nm (262 nm for hypro and proline) using the HP-1100 automatic amino acid analyzer (Agilent, Santa Clara, CA, USA). The levels of amino acids were quantitatively analyzed using a calibration curve constructed with known concentrations of a standard (A9781 from Sigma–Aldrich, Burlington, MA, USA, 0.5 µmol/mL) [36].

### 2.4. Analysis of Elements

The materials, with 10 g of each per sample, were digested separately with a 1:4 (*v*/*v*) mixture of HClO_4_ and HNO_3_, as described by Wang et al. [35]. Elements were detected using atomic absorption spectroscopy (Optima 5300DV, PerkinElmer, Waltham, MA, USA), except for Se, Pb, and Hg, which were analyzed with atomic fluorescence spectrometry (AFS-230E, Haiguang Instruments, Beijing, China). Element standard solutions were used to calibrate the analyzer and to calculate the content of elements in the samples.

### 2.5. Statistical Analysis

All experiments were conducted in triplicate, and statistical significance was determined using SPSS 17.0 (SPSS Inc., Chicago, IL, USA). Results are presented as mean ± standard deviation (SD). Significant differences among groups were assessed using Duncan’s multiple range test, with significance indicated at the 0.05 level by different lowercase letters [37].

## 3. Results

### 3.1. The Idiomorphs and the Fruiting Bodies

*O. xuefengensis* has so far been discovered exclusively in Dongkou County and Wugang City within Hunan, China, at elevations exceeding 600 m in the Xuefeng Mountain region. During cultivation, the presence of two mating types of mycelium simultaneously results in sparse and low-yielding fruiting bodies in solid culture. In this experiment, mononucleate hyphae were selected using DAPI fluorescence staining. On 1% PDA medium, the colonies of *O. xuefengensis* exhibited circular, powdery, and dense morphology, with branched, septate hyphae measuring 2 to 5 µm in diameter. Conidiophores were solitary, producing bottle-shaped structures tapering to a narrow neck, and bearing solitary conidia that were elliptical or elongated, slightly pointed at the apex, measuring (8 to 12) µm × (5 to 7) µm (see Figure 1A,B). The stromata, inoculated onto fruiting bodies culture medium, were cylindrical, with diameters ranging from 1 to 6 mm and lengths of up to 130 mm (Figure 1C,D). The growing tip was beige, and as growth progressed, the outer wall of the stroma gradually changed to yellow-brown while the interior remained beige, displaying a hard texture consistent with natural *O. xuefengensis* stromata.

### 3.2. Specificity of LAMP

Because it is difficult to distinguish between the *MAT1-1* and *MAT1-2* strains based on the morphology, a real-time molecular identification assay using LAMP technology was designed. The results of the LAMP reaction were visualized using HNB dye staining, and the reaction products were analyzed via 2% agarose gel electrophoresis (Figure 2). The positive reaction mixtures stained with HNB appeared sky-blue, and the gel electrophoresis revealed a characteristic DNA ladder of different sizes, displaying a reproducible ladder-like pattern. In contrast, the negative reactions remained violet, and DNA laddering were undetectable after electrophoresis. We further tested a range of isolated *O. xuefengensis* strains using this assay. A total of 14 strains of *O. xuefengensis* with unknown mating types were evaluated to assess the practicality of LAMP detection. The assay clearly differentiated all 14 *O. xuefengensis* isolates, as shown in Figure 2(A1,B1). The specific primers for *MAT1-1* and *MAT1-2* effectively recognized and distinguished the mating types of *O. xuefengensis*. The genotypes of strains numbered 1–4, 8, 9, and 12 are MAT1-1, while strains numbered 5–7, 10, 11, 13, and 14 are MAT1-2. Overall, both the chromogenic reaction and agarose gel electrophoresis were consistent.

### 3.3. Carbohydrates, Nucleosides, and Ergosterol

The quantities of cordycepic acid, total polysaccharides, eight nucleosides, and ergosterol in the various fruiting bodies of *O. xuefengensis* were illustrated in Figure 3 and detailed in Table 2, with results presented as mean ± standard deviation. It is evident from Table 2 that there were statistically significant differences in cordycepic acid, with the *MAT1-2* fruiting bodies showing a slight predominance. The cordycepic acid contents in three samples were 6.7%, 11.7%, and 7.8%, respectively. No statistically significant difference in polysaccharide content was found between the *MAT1-1* fruiting bodies and the *MAT1-2* fruiting bodies. The concentration of adenosine in *MAT1-1* fruiting bodies was 1305 ± 56 mg/kg, which was obviously higher than *MAT1-2* fruiting bodies (649 ± 14 mg/kg) and natural fruiting bodies (762 ± 56 mg/kg). For thymidine, guanosine, adenine, inosine, uracil, and uridine, the following observations were made: thymidine was detected only in the cultured fruiting bodies, while inosine was found exclusively in the natural fruiting bodies. The highest concentrations of adenine, guanosine, and uridine were found in *MAT1-1* fruiting bodies, which contained 256.5 ± 8.2 mg/kg, 2395 ± 40 mg/kg, and 2336 ± 40 mg/kg, respectively. The uracil content in the *MAT1-2* fruiting bodies was 147.4 ± 6.4 mg/kg, which was higher compared to that in *MAT1-2* fruiting bodies and natural fruiting bodies. Except for uracil, the *MAT1-1* fruiting bodies had significantly higher amounts of adenosine, thymidine, guanosine, adenine, and uridine compared to the *MAT1-2* fruiting bodies. Thus, the *MAT1-1* fruiting bodies of *O. xuefengensis* may be particularly favorable for further studies on pharmacological activity of this species and may have potential applications in the development of health and functional medical products.

Ergosterol, a type of mycosterol abundantly present in the cell membranes of fungi, performs a multitude of functions similar to those of cholesterol in animal cells. This study revealed that the ergosterol content in the *MAT1-2* fruiting bodies of *O. xuefengensis* was higher, at 7116 ± 151 mg/kg, compared to 4660 ± 88 mg/kg in the *MAT1-1* fruiting bodies and 5308 ± 10 mg/kg in the natural fruiting bodies.

### 3.4. Amino Acid Composition

The amino acid compositions of different fruiting bodies of *O. xuefengensis* are given in Table 3. Significant discrepancies in amino acid components among the three samples were observed. The contents of total amino acids and total essential amino acids in the *MAT1-1* fruiting bodies were 270.3 ± 2.9 mg/g and 102.5 ± 1.5 mg/g, respectively, which were significantly higher than those in the *MAT1-2* fruiting bodies. The abundance of individual amino acids in the *MAT1-1* sample was much higher than in the *MAT1-2* sample, except for three amino acids: histidine, proline, and arginine. Aspartic acid was the principal amino acid in all three samples of *O. xuefengensis*. The four principal amino acids and their amounts in the *MAT1-1* fruiting bodies were as follows: glutamic acid (38.0 ± 1.7 mg/g), aspartic acid (31.4 ± 1.4 mg/g), leucine (26.3 ± 0.4 mg/g), and arginine (25.4 ± 2.0 mg/g). However, their levels in the *MAT1-2* fruiting bodies and the natural fruiting bodies differed: glutamic acid (21.2 ± 1.0 mg/g and 15.2 ± 1.0 mg/g), aspartic acid (27.8 ± 1.2 mg/g and 20.0 ± 1.2 mg/g), leucine (11.1 ± 0.3 mg/g and 19.0 ± 0.9 mg/g), and arginine (25.5 ± 0.6 mg/g and 15.3 ± 0.6 mg/g), respectively. In the natural fruiting bodies, the total amino acid (TAA) content was 211.55 mg/g, which was notably lower than that in the *MAT1-1* fruiting bodies. There were no notable differences in the content of total essential amino acids (TEAs) between the *MAT1-1* fruiting bodies and natural fruiting bodies. These data indicate that the cultured *MAT1-1* fruiting bodies have a better amino acid composition than the natural fruiting bodies and may be considered as a potential substitute for natural *O. xuefengensis*.

### 3.5. Elements

According to their relationship to human health, elements can be categorized into three groups: essential trace elements, possibly essential elements, and potentially toxic elements. Essential trace elements include iron (Fe), zinc (Zn), copper (Cu), cobalt (Co), selenium (Se), strontium (Sr), molybdenum (Mo), and chromium (Cr), etc. Possibly essential elements include manganese (Mn), nickel (Ni), vanadium (V), and barium (Ba), etc. Potentially toxic elements include arsenic (As), lead (Pb), mercury (Hg), cadmium (Cd), aluminum (Al), and tin (Sn), etc. The average concentrations of these elements, expressed in milligram per kilograms of dry weight in samples, are presented in Table 4.

The concentrations of two essential elements, Cu and Zn, in the cultured *MAT1-2* fruiting bodies were noticeably higher than those in *MAT1-1* and natural fruiting bodies. Selenium (Se) is an important essential microelement known for its antioxidant properties and its role in thyroid hormone metabolism, immune responses, and reproduction in humans and animals. Among the different samples, the Se content in *MAT1-1* fruiting bodies was the highest, reaching 0.43 ± 0.02 mg/kg. Regarding possibly essential elements, the concentration of Mn in *MAT1-1* fruiting bodies (9.52 ± 0.46 mg/kg) was slightly higher than that in the *MAT1-2* fruiting bodies (6.33 ± 0.31 mg/kg). Among potential toxicity elements, the *MAT1-1* fruiting bodies had lower levels of Pb compared to the other two samples, and no significant differences in As levels were observed among the three samples. Overall, *MAT1-1* fruiting bodies exhibited lower levels of toxic elements.

## 4. Discussion

In heterothallic ascomycetes, a single-spore isolate contains either the *MAT1-1* or the *MAT1-2* idiomorph [38]. Currently, the *MAT1-1* and *MAT1-2* strains have identical ITS sequences and can only be distinguished via PCR amplification of mating type genes. In our research, we designed a LAMP assay to rapidly differentiate between the two idiomorphs of *O. xuefengensis*. This is also the first time LAMP technology has been employed to distinguish between the two idiomorphs of *Cordyceps sensu lato.* Compared to PCR, LAMP displayed several advantages, including a short duration (1 h or less), visual detection, and the elimination of the need of a thermal cycler and gel electrophoresis. This technology has proven effective for precise species differentiation. For instance, Blaser et al. [31] distinguished *Bemisia tabaci* from other invasive pests using LAMP, based on the mitochondrial COI gene. He et al. [29] developed a LAMP assay to distinguish lethal *Amanita* species from the other *Amanita* species, using the section *Phalloideae* as the out-group. Li et al. [30] optimized the LAMP detection procedure for *Phytophthora hibernalis*, *P. cambivora,* and *P. syringae* by designing specific primers based on the sequences of internal transcribed spacers (ITS), enolase (*Enol*), and ras-like protein *Ypt1* genes. Zhao and Feng [28] developed a LAMP assay for detecting the Zika virus using the envelope protein (EP) coding region and the NS5 protein coding region as target sequences.

Comparative research was conducted to assess the chemical composition of the fruiting bodies of two mating-type idiomorphs of cultured *O. xuefengensis* and natural *O. xuefengensis*. The cordycepic acid levels in the three fruiting bodies were found to be comparable to those reported for *O. sinensis* by Wang et al. [35], ranging from approximately 6.7% to 11.7% and 4.8% to 11.5%, respectively. The polysaccharide content in the fruiting bodies of *O. xuefengensis* (10.33–12.2%) was slightly higher than the values (4.2–10.43%) reported for natural and cultured *O. sinensis* by Li et al. [39] and Wang et al. [35]. The ergosterol content in the three fruiting bodies of *O. xuefengensis* (ranging from 4.66 mg/g to 7.11 mg/g) was markedly higher than the value reported for cultivated mushrooms of *C. militaris* (3.461 mg/g) by Hu et al. [40] as well as the range observed in wild *O. sinensis* (0.91 mg/g to 4.19 mg/g) by Li et al. [41]. The amounts of five nucleosides (adenosine, thymidine, adenine, guanosine, and uridine) as well as the total essential amino acids and total amino acids in the *MAT1-1* fruiting bodies of *O. xuefengensis* were significantly higher than those in the *MAT1-2* fruiting bodies of *O. xuefengensis* and in natural *O. sinensis* [35,42,43,44]. Cordycepin was first identified in *C*. *militaris* in 1964 and subsequently attracted widespread attention due to its antioxidant activity and anticancer activity [45]. However, recent studies have raised controversy over whether *O. sinensis* and *O. xuefengensis* contained cordycepin [21,22,24]. In our test, cordycepin was undetectable in both *MAT1-1* and *MAT1-2 O. xuefengensis*.

Wang et al. [34] found that the content of toxicity elements in *O. sinensis* was somewhat above the standard, particularly for Al, Pb, As, and Hg. Remarkably, the concentrations of toxic elements (Al, Pb, and Hg) in the *MAT1-1* fruiting bodies of *O. xuefengensis* were undetectable. These data suggest that the *MAT1-1* fruiting bodies of *O. xuefengensis* could be a potential substitute for natural *O. sinensis*. Excessive levels of toxic elements in medicinal and edible fungi have long been a serious problem, becoming a main factor limiting the development of their production. Therefore, selecting specific mating type strains may offer a new solution.

## 5. Conclusions

In the present study, the specific designed nested primer sets allowed for rapid and sensitive differentiation of the two *MAT* idiomorphs of *O. xuefengensis* using the LAMP reaction, providing a new technique for distinguishing different genotypes of the same species. The contents of active constituents in *MAT1-1 O. xuefengensis* were significantly higher than those in *MAT1-2*. More importantly, the contents of heavy metals in the *MAT1-1* fruiting bodies were within safe limits. Altogether, the findings of this study suggest that *MAT1-1 O. xuefengensis* might be a potential substitute for natural *O. sinensis*. Currently, the exploitation and utilization of these species are severely restricted due to the relatively fragile ecological environment of their habitats, limited wild resources, excessive heavy metal content, and high cultivation difficulty. The artificial cultivation of *O. sinensis* fruiting bodies offers a sustainable alternative to harvesting wild specimens, thereby supporting ecological conservation. This approach addresses the scarcity of natural resources and marks a significant advancement toward industrial-scale production of *Cordyceps.* Based on the practical needs for the safe production and green development of *Cordyceps* medicinal materials, our research offers new solutions to the challenges of difficult, lengthy, and unstable artificial cultivation of *Cordyceps* fruiting bodies, as well as the issue of excessive heavy metal content.

## Figures and Tables

**Figure 1 biology-13-00686-f001:**
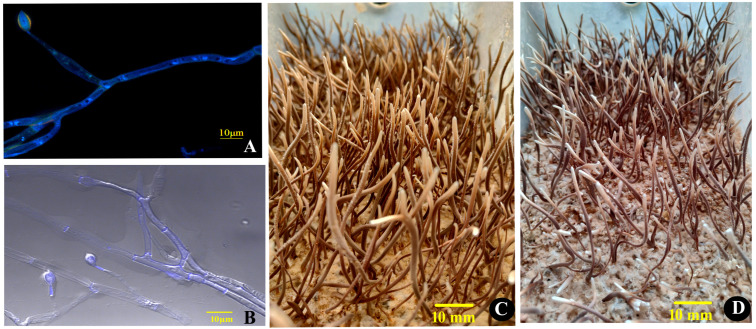
*Ophiocordyceps xuefengensis*. Micrographs of *MAT1-1* hyphae (**A**) and *MAT1-2* hyphae (**B**); macrographs of *MAT1-1* fruiting bodies (**C**) and *MAT1-2* fruiting bodies (**D**). Scale bars: (**A**,**B**) = 10 µm; (**C**,**D**) = 10 mm.

**Figure 2 biology-13-00686-f002:**
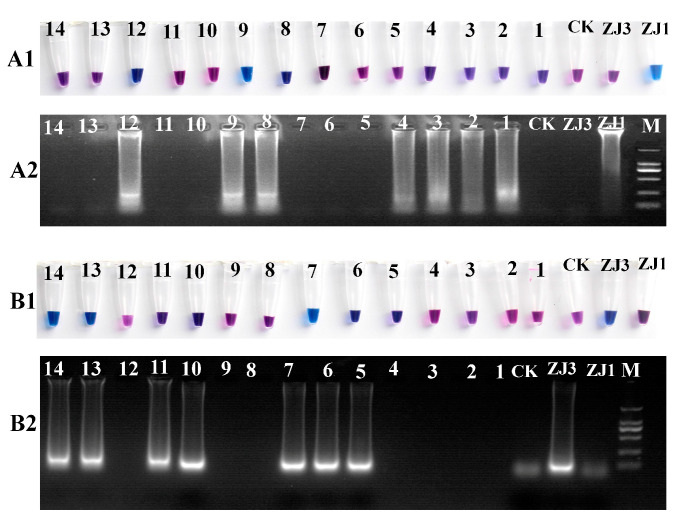
LAMP assay reactions of *O. xuefengensis. (***A1**,**B1**) visual inspection of LAMP products; (**A2**,**B2**) gel images showing LAMP products; (**A1**,**A2**) specific primers for *MAT1-1*; (**B1**,**B2**) specific primers for *MAT1-2*; M: DL2000; ZJ1: *MAT1-1* strain; ZJ3: *MAT1-2* strain; CK: negative control (no DNA); 1–14, 14 strains with unknown mating types.

**Figure 3 biology-13-00686-f003:**
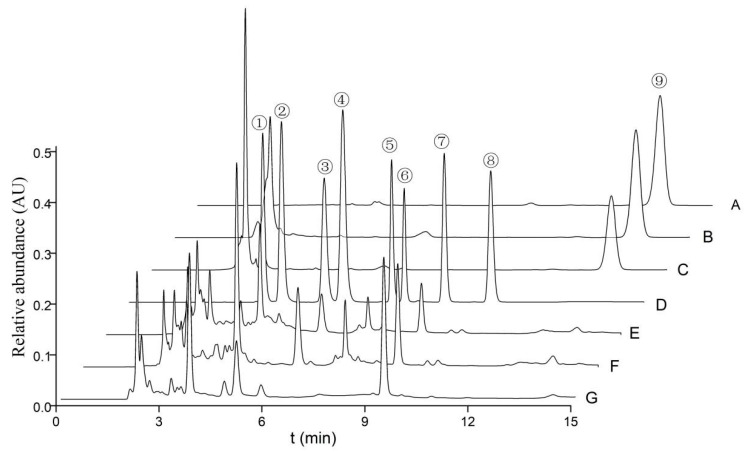
HPLC profile of nucleosides, nucleobases, and ergosterols from *O. xuefengensis*. A–C: Ergosterol (A. standard sample; B. MAT1-2 fruiting bodies; C. MAT1-1 fruiting bodies); D–G: nucleosides and nucleobases (D. standard sample; E. MAT1-2 fruiting bodies; F. MAT1-1 fruiting bodies; G. natural fruiting bodies). 1. uracil; 2. uridine; 3. inosine; 4. guanosine; 5. adenine; 6. thymidine; 7. adenosine; 8. cordycepin; and 9. ergosterol.

**Table 1 biology-13-00686-t001:** LAMP target genes and primers used for amplification.

Target Gene	Primer Name	Length (bp)	Sequence(5′-3′)
*MAT1-1-1*	F3-M1	24	GTGGCTGACAATAAAACTGTAGGT
	FIP-M1	46	CCCACTTGTTTCTGAACGGGTCTTtaatTTCCCGACGTGCAGCAAA
	BIP-M1	46	TGGCAAAGACAAAGTTTCTCTGGCtataGCAGCGGGCTCGATGAT
	B3-M1	17	CCCAGCGCGTTCAGGTA
*MAT1-2-1*	F3-M2	20	CAACATGAACCCCAATCCTC
	FIP-M2	45	AATCCCCCTCCAGGCAAAGAACtaatACGAGGCAATCTGGAAAGG
	BIP-M2	50	TCATGTGAGTCTGATGTTAACCGCAtataTGACTCCTGAACATGCTCCCT
	B3-M2	16	CCCGGTCGGGTCCATT

**Table 2 biology-13-00686-t002:** Contents of cordycepic acid, 8 nucleosides, and ergosterol in the fruiting bodies of *MAT1-1* and *MAT1-2* of *O. xuefengensis*.

Analyte	The Fruiting Bodies of *MAT1-1*	The Fruiting Bodies of *MAT1-2*	Natural Fruiting Bodies
Nucleosides (mg/kg)			
Adenine	256.5 ± 8.2 ^a^	105.0 ± 9.2 ^b^	83.7 ± 8.9 ^c^
Adenosine	1305 ± 56 ^a^	649 ± 14 ^c^	762 ± 56 ^b^
Cordycepin	nd	nd	nd
Guanosine	2395 ± 40 ^a^	1017 ± 26 ^b^	514 ± 18 ^c^
Inosine	nd	nd	178 ± 1
Thymidine	148.4 ± 6.6	83.1 ± 2.1	nd
Uridine	2336 ± 40 ^a^	1400 ± 35 ^b^	975 ± 42 ^c^
Uracil	101.7 ± 7.3 ^b^	147.4 ± 6.4 ^a^	69.1 ± 2.7 ^c^
Carbohydrates (%)			
Cordycepic acid (Mannose)	6.7 ± 0.2 ^c^	11.7 ± 0.7 ^a^	7.8 ± 0.5 ^b^
Polysaccharides	11.8 ± 0.6 ^a^	12.2 ± 0.8 ^a^	10.0 ± 0.6 ^b^
Ergosterol (mg/kg)	4660 ± 88 ^c^	7116 ± 151 ^a^	5308 ± 10 ^b^

Values followed by different letters within the same row indicate significant differences at *p* < 0.05 according to Duncan’s multiple range test; nd, not detected.

**Table 3 biology-13-00686-t003:** Amino acid composition of the fruiting bodies of *O. xuefengensis*.

Amino Acid (mg/g)	*MAT1-1*	*MAT1-2*	Natural Fruiting Bodies
Ile *	14.4 ± 0.2 ^a^	6.8 ± 0.3 ^c^	13.0 ± 1.0 ^b^
Leu *	26.3 ± 0.4 ^a^	11.1 ± 0.3 ^c^	19.0 ± 0.5 ^b^
Lys *	11.7 ± 0.2 ^b^	10.2 ± 0.4 ^c^	15.6 ± 0.8 ^a^
Met *	3.7 ± 0.2 ^b^	2.7 ± 0.2 ^c^	9.4 ± 0.6 ^a^
Phe *	12.3 ± 0.8 ^b^	9.2 ± 0.8 ^c^	19.8 ± 0.9 ^a^
Thr *	10.8 ± 0.7 ^a^	7.9 ± 0.4 ^b^	7.7 ± 0.3 ^b^
Trp *	11.9 ± 1.1 ^a^	2.6 ± 0.3 ^b^	3.1 ± 0.2 ^b^
Val *	11.4 ± 0.7 ^b^	10.5 ± 0.6 ^b^	17.2 ± 0.4 ^a^
Ala	10.5 ± 0.6 ^a^	8.8 ± 0.3 ^b^	11.0 ± 1.0 ^a^
Arg	25.4 ± 2.0 ^a^	25.5 ± 0.6 ^a^	15.3 ± 0.6 ^b^
Asp	31.4 ± 1.4 ^a^	27.7 ± 1.2 ^b^	20.0 ± 1.2 ^c^
Cys	4.7 ± 0.3 ^a^	1.8 ± 0.2 ^c^	3.9 ± 0.3 ^b^
Glu	38.0 ± 1.7 ^a^	21.2 ± 1.1 ^b^	15.2 ± 0.9 ^c^
Gly	12.7 ± 0.5 ^a^	8.0 ± 0.4 ^b^	8.6 ± 0.4 ^b^
His	12.3 ± 1.3 ^b^	14.6 ± 0.4 ^a^	8.6 ± 0.5 ^c^
Pro	10.4 ± 0.6 ^b^	13.0 ± 0.9 ^a^	11.9 ± 0.4 ^a^
Ser	11.7 ± 0.3 ^a^	6.1 ± 0.3 ^b^	6.8 ± 0.5 ^b^
Tyr	10.7 ± 0.9 ^a^	3.9 ± 0.1 ^c^	5.4 ± 0.4 ^b^
Total amino acids (TAAs)	270.3 ± 2.9 ^a^	191.5 ± 2.8 ^c^	211.5 ± 1.7 ^b^
Total essential amino acids (TEAs)	102.5 ± 1.5 ^a^	60.8 ± 2.0 ^b^	104.9 ± 1.0 ^a^
TEA/TAA (%)	37.9	31.7	49.6

Values followed by different letters within the same row indicate significant differences at *p* < 0.05 according to Duncan’s multiple range test; * indicates an essential amino acid.

**Table 4 biology-13-00686-t004:** Elemental composition of natural and cultured fruiting bodies of *O. xuefengensis* (mg/kg).

Minerals	*MAT1-1*	*MAT1-2*	Natural Fruiting Bodies
Ca	1222 ± 77 ^b^	1200 ± 61 ^b^	4900 ± 139 ^a^
Mg	887.33 ± 34.85 ^c^	1300 ± 45.18 ^a^	1000 ± 19.08 ^b^
K	10763 ± 1292 ^b^	14967 ± 1320 ^a^	7800 ± 300 ^c^
Na	203.33 ± 14.98 ^b^	147 ± 9.85 ^b^	6000 ± 148.39 ^a^
Fe	10.77 ± 0.61 ^b^	9.85 ± 0.62 ^c^	59.34 ± 2.23 ^a^
Zn	71.00 ± 4.58 ^b^	142.8 ± 8.0 ^a^	43.86 ± 2.88 ^c^
Cu	12.12 ± 1.65 ^b^	16.75 ± 2.25 ^a^	6.84 ± 0.24 ^c^
Co	nd	nd	nd
Se	0.43 ± 0.02 ^a^	0.14 ± 0.01 ^c^	0.35 ± 0.01 ^b^
Sr	0.78 ± 0.02	nd	nd
Mo	nd	nd	nd
Cr	0.76 ± 0.04 ^c^	1.87 ± 0.03 ^a^	0.85 ± 0.05 ^b^
Mn	9.52 ± 0.46 ^b^	6.33 ± 0.31 ^c^	23.99 ± 1.32 ^a^
Ni	nd	nd	nd
V	nd	nd	nd
Ba	nd	nd	nd
As	0.53 ± 0.04 ^a^	0.54 ± 0.02 ^a^	0.53 ± 0.03 ^a^
Pb	nd	2.84 ± 0.07	0.40 ± 0.02
Hg	nd	nd	nd
Cd	nd	nd	nd
Sn	nd	1.23 ± 0.06	0.26 ± 0.01
Al	nd	nd	nd

Values followed by different letters within the same row indicate significant differences at *p* < 0.05, according to Duncan’s multiple range test; nd, not detected.

## Data Availability

Data will be made available on request.

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
