# Peer review of "MAT1-1 and MAT1-2 Ophiocordyceps xuefengensis and Comparison of Their Chemical Composition"

_biology, 2024, doi:10.3390/biology13090686_

Round 1
Reviewer 1 Report
Comments and Suggestions for Authors
Just like most basidiomycetes fungi, ascomycetes fungi also have homothallism,heterothallism and pseudohomothallism. The heterothallic ascomycetes fungi is only bipolar, and the more complicated one is pseudohomothallic fungi, they can form normal fruiting bodies. As the species of Cordyceps sensu lato and Morchella which can be cultivated artificially, pseudohomothallic phenomenon exists. In this paper, the authors tried to study the effect of mating type on the chemical composition of pseudohomothallic Ophiocordyceps xuefengensis. But the following problems were obviously existed in paper.
1. Although the idiomorph of the O. xuefengensis isolate can be rapid identified by using loop-mediated isothermal amplification (LAMP), the identification is qualitative rather than quantitative. So the detection sensitivity should be explored using the genomic DNA template of the positive strain. Sample 7 in Figure 2A1 is slightly negative, and the sample 7 is not significantly different in color from samples1, 2, 3, and 4.
2. Primers and PCR amplification procedures used for the identification of idiomorphs should be added in the methods.
3. In figure 1, it is need to show that whether the normal ascospores can be found in the MAT1-1951 fruiting bodies and the MAT1-2 fruiting bodies. The ascocarps were compared with those of the parent strains.
4. 4. As a statistical analysis, the smple numbers are more than three strains and three cultivation repeats, otherwise there is no statistical significance.
5. All test materials in this study should be listed in the material.
6. In paper, all body should be fruiting bodies.
7. In paper, "the idiomorph" was used as MAT1-1 1-1 strain or MAT1-2-1 strain, e.g fruiting bodies between the idiomorphs, that's wrong.

Reviewer 2 Report
Comments and Suggestions for Authors
The work was carried out at a good methodological level, using the technology of genotyping mating-type fungi and determining the most important metabolites in them. However, the presentation is not clear; in particular, the authors should indicate which strains of Ophiocordyceps xuefengensis they worked with, and provide information either on the collection number or on genotyping data for new isolates. It is also not completely clear why it was necessary to use the LAMP reaction technology specifically in this work, and not use traditional PCR amplification to determine the type of mating. The work itself does not seem to illustrate the use of the LAMP reaction, since it is devoted to “MAT1-1 and MAT1-2 Ophiocordyceps xuefengensis and comparison of their chemical composition.” The development of an express LAMP reaction method seems to be an additional task after traditional PCR diagnostics. The authors need to explain more clearly why they immediately and only used the LAMP reaction. The authors also need to more clearly indicate how many MAT1-1 and MAT1-2 strains they characterized in terms of metabolites and elemental contents (Tables 2-4), conduct statistical analysis to prove that such a number of individual strains gives statistically reliable results about the difference in meaning. For a clearer understanding of the processes of switching the mating type in Ophiocordyceps xuefengensis, it is highly desirable to add an additional figure with scheme (or currently assumed scheme) of the life cycle of Ophiocordyceps xuefengensis. The work deserves publication in the journal Biology after correction in accordance with the comments made.
Comments
The Instructions for Authors state that “A simple summary consists of no more than 200 words.” In the current version, the abstract consists of 358 words, which significantly exceeds the established limit. Reformulate the abstract to the required 200 words, focusing only on the main results.
Lines 12-13.
Reformulate this sentence, in the current version it turns out that Cordyceps are traditional Chinese medicinal herbs, although Cordyceps belong to fungi, the class of Soradiomycetes.
Lines 65-67
Check out this statement. Ergosterol contains four linked hydrocarbon rings. In the current version you only list three of the four rings.
Table 1.
Add a separator line to make it clear which primers were used to identify MAT1-1-1 and which were used for MAT1-2-1. This must be done because, In the current version of the table, for example: primer F3 can be classified as both MAT1-1-1 and MAT1-2-1.
Rename the primer names so that different sequences are not referred to by the same abbreviation. In the current design you have a 24 bp F3 primer and a 20 bp F3 primer, these primers are not only different sizes but have no significant homology. And so for all primers. Correct according to this Lines 141-142.
Lines 110-125
Indicate the source of O. xuefengensis strain(s). If you took them from a collection of microorganisms, please provide the accession number. In the current version of the article you write “The fungal strains, no. 1-14, respectively, were isolated from the fresh stromata of O. xuefengensis gleaned from the Xuefeng Mountains in China." If you yourself received this isolate(s), then provide information about which barcoders you used to genotype this strain(s), and also provide an additional Figure with sampling map (with the China region) in your article.
Also indicate the source of the ZJ1 and ZJ2 strains you used as reference strains for types MAT1-1 and MAT1-2 (Figure 2).
Write in more detail for how long and on what medium you carried out the cultivation to obtain fruiting bodies
Lines 126-147
Explain the molecular basis on which you differentiate between MAT1-1 and MAT1-2 strains of O. xuefengensis. In the MAT1-1 strain, does O. xuefengensis completely lack genetic information for the MAT1-2 locus, or is it located in the heterochromatin region (just as MAT1-2 does not have genetic information for MAT1-1)? This is important to indicate because many heterothallic fungi have both information about mating types at the same time. For example, in S. cerevisiae, the genomes of both MATa and MATα mating type strains contain information about both loci, which is stored as heterochromatin at the end of the chromosome III. However, the centromeric zone of the chromosome III also contains information about the current mating type in the form of euchromatin. Thus, MATa strains have two MATa and one MATα loci. Mating type can potentially be determined by gene dosage. As in the case of O. xuefengensis, the MAT1-1 strains also have the MAT1-2 locus (but at a lower gene dose), and are you using a LAMP reaction rather than PCR (or qPCR) to more accurately differentiate the higher gene dose at the target locus? Write about switching the mating type and the possibility of molecular diagnostics of these heterothallic strains of O. xuefengensis in any section (introduction, materials and methods, results and discussion, or in the discussion).
Lines 170-176
Perhaps this section would be better called elements, since you did not detect minerals, but elemental composition.
Write in this section, in which samples did you determine the elements, in the original isolates, or in the cultured samples? If you determined the content of cultured samples, then provide a description of the composition of the medium, in particular, in what concentrations of heavy metals were added during cultivation.
Figure 1.
Write in the caption to the Figure what method you used to obtain images A, B and C, D. In the materials and methods section, add a description of this method and write what device was used to obtain these images. For Figure 1 C,D, add a scale bar.
Figure 3.
What is meant by "G. Natural fruit body” - then what is this mixture of MAT1-1 and MAT1-2? If so, what better place to write "G. Natural fruit body (mix of MAT1-1 and MAT1-2) ”
Indicate in Tables 2-4 for what number of strains MAT1-1, MAT1-2 and Natural fruit body the average values ​​are given.
Reviewer 3 Report
Comments and Suggestions for Authors
The work MAT1-1 and MAT1-2 Ophiocordyceps xuefengensis and compar- 2 ison of their chemical composition, described the efficiently and rapidly distinguish the idiomorphs of Ophiocordyceps xuefengensis, while systematically compare various biochemical ontents of fruiting bodies between the idiomorphs.
The work is well written, just small suggestions:
1- The authors could make the novelty of the work clear.
2- HPLC analysis method validation data could be inserted.
Reviewer 4 Report
Comments and Suggestions for Authors
The research topic is of great medical importance, but the results of the study are controversial. It is necessary to re-conduct the experiments, increasing the repetition to 5 times. This will allow you to compare the data more accurately. If the sample was contaminated, 5 different samples must be re-taken and analyzed again (not a 5-fold chemical analysis of one sample, but a one-time analysis of 5 different samples).
Repeated analyzes and their correct statistical processing will allow both the authors and readers of the article to be convinced of the correctness of the results.
Specific shortcomings of the manuscript.
1. Line 12: “medicinal herbs”?
2. Line 50, 52, 75, 76, 78 and some others: you need to write the full name.
3. Line 102-107: This passage is not worded in the style of the Introduction section. It is more in line with the style of Material and Methods. Rewrite again.
4. Line 125: you need to clearly indicate the coordinates.
5. Latin names must be italicized (for example, line 112, 154).
6. Statistical processing should be described in detail in subsection 2.5.
7. The Results section cannot contain references to literature and proposals that compare your own results with the results of other authors. This should be moved to Discussion.
8. The rounding of numbers is unsuccessful (for example, line 233, 242, 252). Excessive detailing of numbers (for example, not to tenths, but to hundredths) indicates that the authors do not understand all the factors that can affect the results of research. In some cases, it is advisable to round to integers (for example, lines 252-253).
9. It is not clear what result is indicated after the number +- (for example, line 242, 252)? Could it be standard error, or standard deviation, or 1.96*standard deviation? Authors should clearly discuss this in subsection 2.5 and repeat it in the titles of figures and tables.
10. For Table 2, three repetitions are not enough. It is necessary to apply fivefold repetition. The authors did not indicate a specific method for comparing samples under the table. The numbers in the table in the lines “Thymidine”, “Adenine” and “Uracil” should be rounded to the nearest tenth. Numbers for the strings "Adenosine", "Guanosine", "Uridine" and "Ergosterol" must be rounded to the nearest whole number.
11. For some reason, the units of measurement in Table 2 are indicated for the first and last rows (and the same units). Because the units of measurement are not the same throughout the table, they must be listed in parentheses in each cell in the first column.
12. Why is the data for “Cordycepic acid (Mannitol)” combined in one row in Table 2? It is not correct.
13. Why is “Polysaccharide” written in the singular in Table 2? It is not an individual substance, but a mixture.
14. Why do the authors believe (Table 2) that “Ergosterol” refers to “Carbohydrates”?
15. The letter designations near the numbers in Table 2 are not clear (line 263, 264). Here it is necessary to use one level of significance of differences (0.05). Statistical analysis requires in-depth testing.
16. I do not believe the numbers in Table 3 in the column “MAT1-1”. Hundredths of values ​​are always equal to "0". This is data falsification. The hundredths must be 0, 1, 2, ....8 and 9 with equal frequency within each column of the table.
17. Most of the comments on Table 2 also apply to Tables 3 and 4.
18. The rounding of numbers in Table 4 is very careless: to whole, tenths and hundredths (without any logic). Carelessness in digital data indirectly indicates carelessness during laboratory research.
19. Why does the iron content differ by more than 10 times (Table 4)? The authors should re-analyze. The sample was probably contaminated. Similar suspicions for calcium and potassium. The aluminum content differs hundreds of times. This is definitely contamination of the sample with clay. It is unacceptable to publish such dubious data. Statistical processing of selenium data (Table 4) is very sloppy. The reason for such questionable data is the low (threefold) replication.
20. What is the logical reason for this arrangement of rows in tables 2-4? Authors have the right to arrange rows according to any logic, but this logic must be specified in a note under each of these tables.
21. There are no DOI indexes in the literature. Commas and periods, as well as extra spaces between initials, are placed carelessly (for example, line 512), not according to the rules of the journal. There is no need to capitalize all words in article titles. Latin names should be italicized (for example, lines 479, 485). Instead of the article number, pages are indicated (for example, line 427).
Round 2
Reviewer 1 Report
Comments and Suggestions for Authors
Authors have good response for commernts. No other comments and sugestions.
Comments on the Quality of English LanguageNO.
Author Response
Thank you for your feedback. The manuscript's language has been revised by Dr. Xiong from Canada. Please see the revised manuscript.
Reviewer 2 Report
Comments and Suggestions for Authors
The authors have done a detailed and thorough job of correcting the results and presenting the material. This has made the work look more complete, especially in its methodological part, and will be easier for the reader to perceive. The authors have responded to all my comments, and I believe that the work can be published in its current form.
Author Response
Thank you !
Reviewer 4 Report
Comments and Suggestions for Authors
This article has improved. However, the authors have made it inaccessible to the reader which of the numbers (in all tables) differs from which. This currently prevents the article from being recommended for publication.
I recommend that the authors carefully read the articles in the world's leading scientific journals and evaluate how other authors use letters to indicate the presence or absence of differences between samples based on the results of multiple comparisons of samples. It is important that readers understand the idea that the authors of the article want to convey to them. What the authors have used letters to indicate in the tables of this article will not be understood by readers. I am sorry.
Author Response
Thank you for your feedback; we have reorganized and simplified the letter annotations in the tables to clearly indicate significant differences among values (P<0.05) based on Duncan's multiple range test. Please see the revised manuscript.